# Implementation of a Commercial Deep Learning-Based Auto Segmentation Software in Radiotherapy: Evaluation of Effectiveness and Impact on Workflow

**DOI:** 10.3390/life12122088

**Published:** 2022-12-13

**Authors:** Lorenzo Radici, Silvia Ferrario, Valeria Casanova Borca, Domenico Cante, Marina Paolini, Cristina Piva, Laura Baratto, Pierfrancesco Franco, Maria Rosa La Porta

**Affiliations:** 1Department of Medical Physics, ASL TO4, Ivrea Community Hospital, 10015 Ivrea, Italy; 2Department of Radiation Oncology, ASL TO4, Ivrea Community Hospital, 10015 Ivrea, Italy; 3Department of Translational Medicine (DIMET), ‘Maggiore della Carità’ University Hospital, University of Eastern Piedmont, 28100 Novara, Italy

**Keywords:** radiation therapy, contouring, auto segmentation, artificial intelligence, delineation

## Abstract

Proper delineation of both target volumes and organs at risk is a crucial step in the radiation therapy workflow. This process is normally carried out manually by medical doctors, hence demanding timewise. To improve efficiency, auto-contouring methods have been proposed. We assessed a specific commercial software to investigate its impact on the radiotherapy workflow on four specific disease sites: head and neck, prostate, breast, and rectum. For the present study, we used a commercial deep learning-based auto-segmentation software, namely Limbus Contour (LC), Version 1.5.0 (Limbus AI Inc., Regina, SK, Canada). The software uses deep convolutional neural network models based on a U-net architecture, specific for each structure. Manual and automatic segmentation were compared on disease-specific organs at risk. Contouring time, geometrical performance (volume variation, Dice Similarity Coefficient—DSC, and center of mass shift), and dosimetric impact (DVH differences) were evaluated. With respect to time savings, the maximum advantage was seen in the setting of head and neck cancer with a 65%-time reduction. The average DSC was 0.72. The best agreement was found for lungs. Good results were highlighted for bladder, heart, and femoral heads. The most relevant dosimetric difference was in the rectal cancer case, where the mean volume covered by the 45 Gy isodose was 10.4 cm^3^ for manual contouring and 289.4 cm^3^ for automatic segmentation. Automatic contouring was able to significantly reduce the time required in the procedure, simplifying the workflow, and reducing interobserver variability. Its implementation was able to improve the radiation therapy workflow in our department.

## 1. Introduction

Radiation therapy (RT) is an important treatment option in the management of cancer. It aims at delivering a high radiation dose to target cancer cells to ensure clinically required tumor control probability and concomitantly spare the nearby healthy tissues to prevent acute RT-related toxicity and late effects.

Accurate contouring of Clinical Target Volumes (CTV) and Organ at Risk (OAR) is important for treatment planning and delivery. Generally, the segmentation of tumor regions and normal tissues is manually performed by the clinical staff, based on the images acquired during planning computed tomography (CT). This approach is prone to a high degree of inter and intra observers’ variability, being time-consuming, and representing a bottleneck in the planning workflow [1].

To improve the efficiency of this process, auto-contouring methods have been proposed. One of the most popular approaches is atlas-based segmentation [2,3]. However, contouring algorithms based on deep-learning techniques are being increasingly used, showing better results than atlas-based approaches [4,5]. The purpose of our study was to investigate the clinical implementation in our institution of a specific deep learning-based auto contour commercial software to assess the impact on the radiotherapy workflow in four specific disease sites: head and neck, prostate, breast and rectum.

## 2. Materials and Methods

### 2.1. Deep Learning Auto-Segmentation

A commercial deep learning-based auto-segmentation software, Limbus Contour (LC), Version 1.5.0 (Limbus AI Inc., Regina, SK, Canada), which uses deep convolutional neural network models based on a U-net architecture specific for each structure, was recently introduced in our institution. The software relies on models trained, using public datasets [6,7,8,9,10], as well as datasets obtained through institutional data agreements [11,12,13,14,15]. The number of scans used in the training set for each model varies: each model is trained on hundreds or thousands of scans. Models were trained using TensorFlow; typical image augmentation and regularization techniques were applied. Each model is validated internally by Limbus AI by comparing the model output on a set of test scans to expert human contours on the same test scans. The models are also validated in published studies that investigate qualitative and quantitative accuracy and time savings [1,16,17,18].

LC obtains information related to the acquisition protocol by reading the DICOM metadata of the CT images. The corresponding auto-segmentation model is then automatically used to create auto-segmented contours that are exported alongside the CT images to the treatment planning software to be eventually edited and then validated by the clinicians.

### 2.2. Patients’ Selection

Four disease sites were selected for the present study, namely Head and Neck (H&N), prostate, rectum, and breast cancer. We focused on these four settings, considering their high frequency and important impact on the radiotherapy workflow. For each type of treatment, three patients treated in our center were selected.

For H&N, we chose oropharyngeal cancer to guarantee the standardization of OARs contouring. Patients eligible for the study received radical radiotherapy. The prescription dose was 70 Gy delivered in 35 fractions for the curative setting. The prostate setting consisted of patients who received exclusive radiotherapy on prostate gland and seminal vesicles. A moderate hypofractionated schedule was proposed: 70 Gy on prostate gland and 63 Gy on seminal vesicles in 28 fractions, delivered with a simultaneous integrated boost. For rectal cancer, patients offered pre-operative RT were considered. The prescription dose was 50 Gy on the gross tumor volume and positive nodes, and 45 Gy on the elective volumes, in 25 fractions. Finally, patients with left-sided breast cancer who underwent conservative surgery were selected. In this case, the prescription dose was 45 Gy for whole breast irradiation and 50 Gy on the tumor bed, given with a concomitant boost, in 20 fractions.

### 2.3. Technical Setup

Each patient underwent a planning CT scan in supine position; to prevent patient’s displacements during treatment, immobilization devices were used: thermoplastic mask for H&N, knee wedge and foot lock for prostate and rectum treatments, breast board for breast cases. Planning CT images were acquired with a Canon Aquilion LB V6.3 series scanner (Canon medical system corporation—Ōtawara, Japan) with 120 kVp tube load. The slice thickness was 3 mm for H&N cancer and 5 mm for other diseases. The in-plane pixel size was 1 mm × 1 mm for all acquisitions.

### 2.4. Contour Methods

Four different RO, each with expertise in the specific clinical setting, manually delineated the four-treatment districts following national and international consensus guidelines [11,12,13,14,15,19,20,21,22]. For contouring purposes, additional imaging (e.g., magnetic resonance imaging, positron emission tomography or diagnostic CT) were used, if necessary. The clinically approved treatment plan was subsequently delivered.

The same CT acquisitions were contoured by LC and the images, together with the RTstructure DICOM file, were then sent to the Treatment Planning System (TPS) Eclipse (Version 15.6, Varian Medical Systems—A Siemens Healthineers Company, Palo Alto, CA, USA). The LC structure set was later duplicated on the TPS. One structure set has been reviewed by the competent RO and, if necessary, the contours were modified; the second was not submitted to any change.

For H&N cancer, contoured OARs were fifteen (brainstem, brachial plexuses, spinal cord, inner ears, parotid glands, thyroid, mandible, oral cavity, larynx, lungs and esophagus). For prostate cases, five structures were considered (bladder, femoral heads, rectum and penile bulb). For rectal cancer, four OARs were accounted for (femoral heads, bladder and bowel—as abdominal cavity). Finally, for breast cancer contoured structures were four (contralateral breast, heart, and both lungs).

### 2.5. Contouring Time

We recorded the time spent performing the manual contour for each CT scan. Moreover, the time required for LC to generate OARs on a consumer grade system (3.1 GHz Intel Core i7, 8 GB memory) was also evaluated. Finally, the time spent by the ROs to review and, if necessary, edit the contours performed by LC was measured. The overall duration of contouring using LC (LC contouring plus ROs review) was compared to the time required to perform manual contouring, which was used as a reference. In this way, the time difference—absolute and relative—between the two contouring methods was obtained.

### 2.6. Geometrical Analysis

The manually contoured structures (MC) were compared with those generated by LC by means of three indicators: volume variation, Dice Similarity Coefficient (DSC) and shift of the center of mass. For structures with a volume greater than 15 cm^3^, the volume percentage variation was considered. Conversely, for smaller structures, the absolute change in volume was analyzed, since the percentage variation was not considered indicative, given that small variations in volume lead to large percentage variations.

DSC [23] is a measure of the overlap of two volumes. Its value is comprised between 0 and 1, where 0 indicates no overlap while 1 stay for complete overlap. If X and Y are the two volumes to be compared, the coefficient DSC (X|Y) is defined as DSC (X|Y) = 2|X∩Y|/(|X| + |Y|). Finally, starting from the coordinates of the center of mass of each structure in latero-lateral (X), cranio-caudal (Y) and antero-posterior (Z) direction, its displacements between manual and auto-segmented contouring were evaluated. All the parameters were obtained from the statistics tool of the contouring module of Eclipse TPS.

### 2.7. Dosimetric Analysis

A dosimetric analysis was performed to evaluate the effects of unsupervised use of LC on the assessment of dose distribution.

The original treatment plan, optimized and clinically approved with the manually contoured volumes, was recalculated on the LC contoured structure-set using the AAA algorithm (version 15.6.06) of Eclipse TPS, the same as the original plan.

The differences in the Dose Volume Histograms (DVH) between the two structure sets were then evaluated and plans were compared using the metrics reported in Table 1.

For serial organs, metrics associated with maximum dose were used, while for parallel organs the average dose or dose too large volume were considered.

## 3. Results

### 3.1. Contouring Time

The absolute and percentage variations of the contouring times are shown in Figure 1. The maximum time saving, both absolute and relative, was obtained for the H&N setting (80 min and 65%, respectively). The minimum changes, both absolute and relative, were found for rectum (3 min and 17%, respectively). Similar variations were found for prostate treatments, while breast cases showed intermediate values.

### 3.2. Geometrical Analysis

Figure 2 shows the average percentage variations in volumes for structures with a volume greater than 15 cm^3^. The associated uncertainty is expressed in terms of ±1 standard deviation. The OAR with the minimum variation (1%) is lung; the structures with the greatest percentage variation are bowel and oral cavity, with mean percentage variations of 65% and 32%, respectively.

The absolute volume variations for structures with a volume smaller than 15 cm^3^ are reported in Figure 3. The associated uncertainty is expressed in terms of ±1 standard deviation. All the structures show values close to or less than 1 cm^3^.

Figure 4 shows the average Dice Index for the analyzed structures, with the relative uncertainty, expressed as ±1 standard deviation. The lowest DSC value is 0.39 for the penile bulb. The best results were found for lungs, characterized by a Dice Index of 0.99. Furthermore, a good agreement was found for bladder, heart, and femoral heads, with values greater than or close to 0.9. Considering all structures, the average DSC is 0.72.

The absolute value of the three-dimensional displacement of the center of mass is represented, for all the structures, in Figure 5. The lowest values were found for lungs, with values close to 0. The greatest displacement occurred for bowel, with a value equal to 2.4 cm. In Figure 6 the absolute values of the displacements in each direction are reported for bowel.

### 3.3. Dosimetric Analysis

In Table 1, the metrics used for the dosimetric comparison of the treatment plans are reported. The most relevant difference was found in the bowel for rectal cancer treatments: the mean volume covered by the 45 Gy isodose was 10.4 cm^3^ for the MC structures versus 289.4 cm^3^ for the LC ones.

## 4. Discussion

The present study explores the effects of commercial deep-learning based software for auto-contouring on the clinical workflow of a radiation oncology department at a tertiary cancer hospital. In particular, the focus was on timesaving and on the accuracy of the contoured structures.

To accurately assess the time reduction, we evaluated the clinical settings having the highest impact on the workflow in our radiotherapy department. In addition, for each disease site we focused on, all the OARs included in the clinical routine were considered.

Limbus performance was already analyzed by other authors, who investigated multi-observer variability [1], qualitative evaluations of expert ROs [21] and specific evaluations for lung SBRT [22,23]. Furthermore, Zabel et al. [16] compared the manual contouring workflow with LC and an additional atlas-based automatic contouring algorithm for bladder and rectum contouring. Finally, a recent study by D’Aviero et al. evaluates the geometric accuracy of the contours limited to H&N district [24]. The present study includes 28 OARs and 4 anatomical subsets, resulting in a total of 84 contours analyzed. To the best of our knowledge, there are no data available in the literature on such a comprehensive list of OARs and diseases. Furthermore, this study investigates the entire radiotherapy workflow, focusing on geometrical accuracy, timesaving and dosimetric implications of LC implementation in a radiotherapy department.

The possibility to save time is greater in anatomical districts characterized by a greater number and complexity of OARs. Our data are similar to those reported in the literature. As an example, in the setting of lung cancer, Lustberg et al. [2] showed an average time saving of 61% compared to existing clinical practice and 22% compared to the use of atlas-based contours. Wong et al. [1] also found remarkable decreases in contouring time, although for H&N the absolute time reduction is not comparable to ours due to the smaller number of structures contoured by Wong et al.

LC provides good results, as no gross contouring errors were found. This high-quality performance is highlighted by the average DICE Index of 0.72 which can be considered acceptable in clinical practice [21]. However, some OARs have characteristics that deserve to be discussed.

As can be seen by center of mass and geometrical analyses, there is a difference in bowel manual contouring versus automatic segmentation. LC considers as bowel the entire abdominal cavity, extending the caudal limit including the whole inferior abdomen, regardless of the presence of the intestinal loops. During manual contour, however, bowel was considered as abdominal cavity whose caudal limit is defined by the presence of intestinal loops [14,25,26]. These differences justify the dosimetric variation observed.

Regarding the oral cavity, the differences are due to different approaches in contouring; similar results are found by Zhong et al. [27]. LC considers the extended oral cavity, as Contouring Head and Neck OARs Guidelines suggest [15], including the oral tongue and anterior portion of the oropharynx. In manual contouring, the latter was instead excluded from the oral cavity OAR, since it is part of PTV.

The low DSC for penile bulb is an expected finding, as the anatomical markers or the necessary soft tissue contrast for the penile bulb is generally lacking on CT. To best identify penile bulbs and reduce great contouring variability, some authors have stressed the importance of performing an MRI or CT scan with contrast in the urethra for optimal identification of the penile bulb [14].

About brachial plexuses, the institutional practice is not to contour the complete brachial plexus until, laterally, the thoracic wall because for oropharynx tumors the dose to the brachial plexus axillary trunk is negligible [28]. This choice is due to the necessity, in manual contouring, to reach a compromise between the contouring time and the usefulness of the executed contour. However, this tradeoff is not necessary in the case of automatic contouring.

A disagreement in the cranial limit of plexuses was also found. During manual contouring, the brachial plexuses start from the spinal nerves through the neural foramina from the C4–C5 (C5 nerve roots) to the T1–T2 (T1 nerve roots) level. In LC the cranial level of brachial plexus is often higher, such as C2–C3, probably due to the position of neck. These issues explain differences in DSC values for brachial plexuses (about 0.7) compared to those found by D’Aviero et al. [24] (about 0.95).

Regarding parotid glands, no significant changes in geometric parameters were found. However, there is a non-negligible variation of dosimetric indicator. Although the shape and position of parotid glands are similar in manual contouring and LC, minimal differences could drastically affect dosimetric parameters because of the proximity of parotid glands to PTV and to the steep dose gradients. These results are similar to those reported by Nelms et al. [29].

Good results were found for lungs, femoral heads and bladder. DSC values for these OARs were similar to those found by Wong et al. for bladder, femoral heads [21] and lungs [22]. Furthermore, Zabel et al. [16] bladder DSC value −0.97—confirms our result.

A limitation of the study is the low number of patients analyzed for each setting. However, the analysis considers all OARs involved in the clinical workflow for the considered anatomical regions. This allows for a comprehensive assessment of the impact of LC on radiotherapy routine and considers all the steps of the radiotherapy planning process, from contouring to dosimetric consequences of the unsupervised use of LC. A complete description of LC impact on radiotherapy routine can provide useful information.

As a novelty, this study provides quantitative evidence of the time savings achieved by LC use. These values are realistic, thanks to the number of contoured structures. Furthermore, it is possible to identify the anatomical sites which most benefit from LC. Dosimetric evaluation shows that, although DVH differences are not significant in most cases, LC contoured structures must always be supervised by an expert contourer. Otherwise, especially in regions near to high dose gradients, there may be relevant dosimetric variations.

## 5. Conclusions

Although an accurate visual review by an expert clinician is still required, LC can significantly reduce the time required for contouring and simplify the workflow leading to treatment planning. Its implementation also allows reducing interobserver variability and improving the interpretation of radiological anatomy. Furthermore, LC can support staff training and the continuous assessment of clinical contouring and structure segmentation. 

## Figures and Tables

**Figure 1 life-12-02088-f001:**
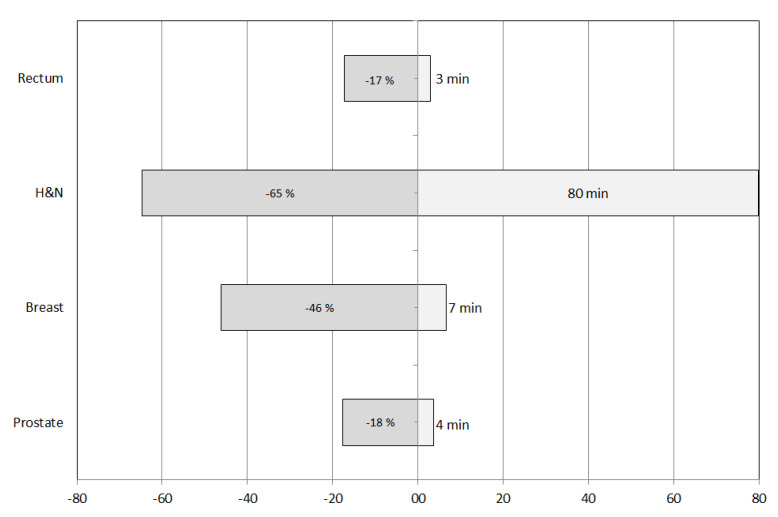
Absolute (**right**) and percentage (**left**) time reduction obtained with Limbus auto-segmentation software.

**Figure 2 life-12-02088-f002:**
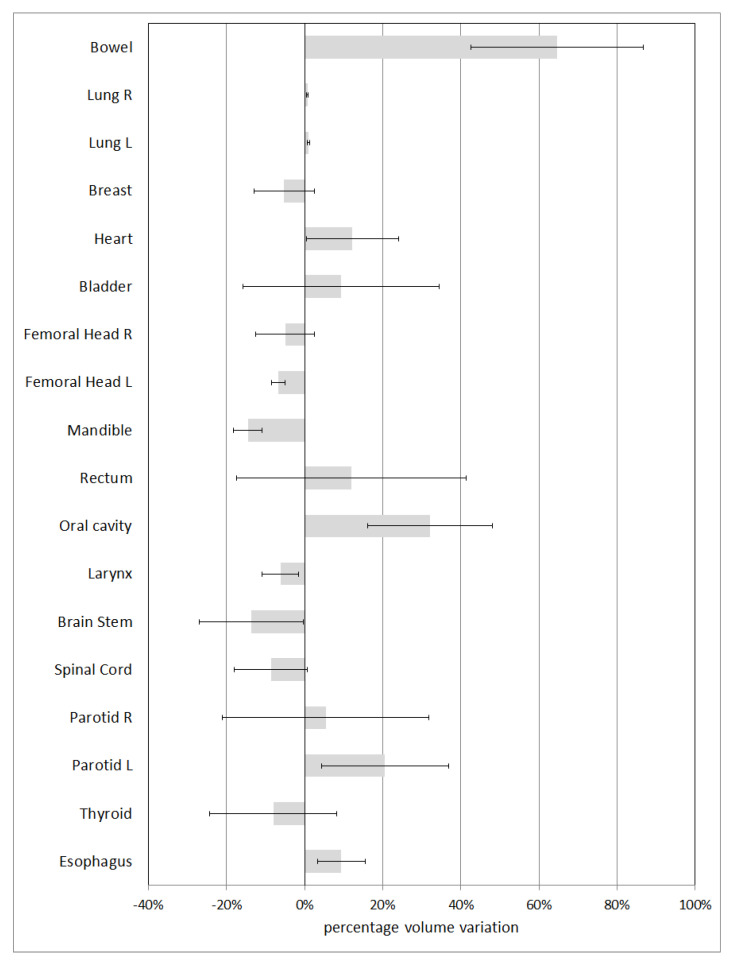
Percentage volume variation for OARs with volume greater than 15 cm^3^.

**Figure 3 life-12-02088-f003:**
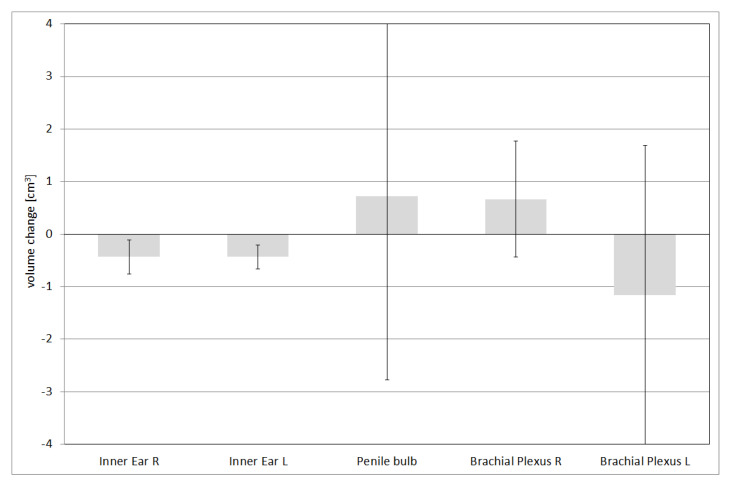
Absolute volume variation for OARs with volume smaller than 15 cm^3^.

**Figure 4 life-12-02088-f004:**
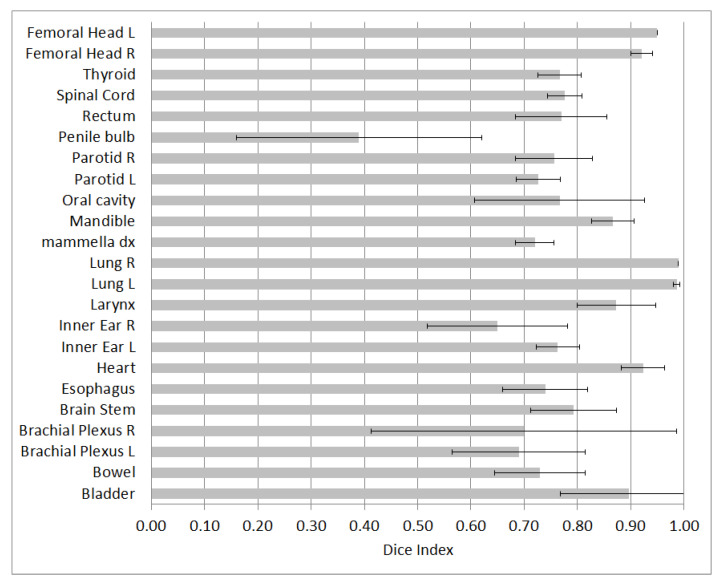
Dice Index for the evaluated structures.

**Figure 5 life-12-02088-f005:**
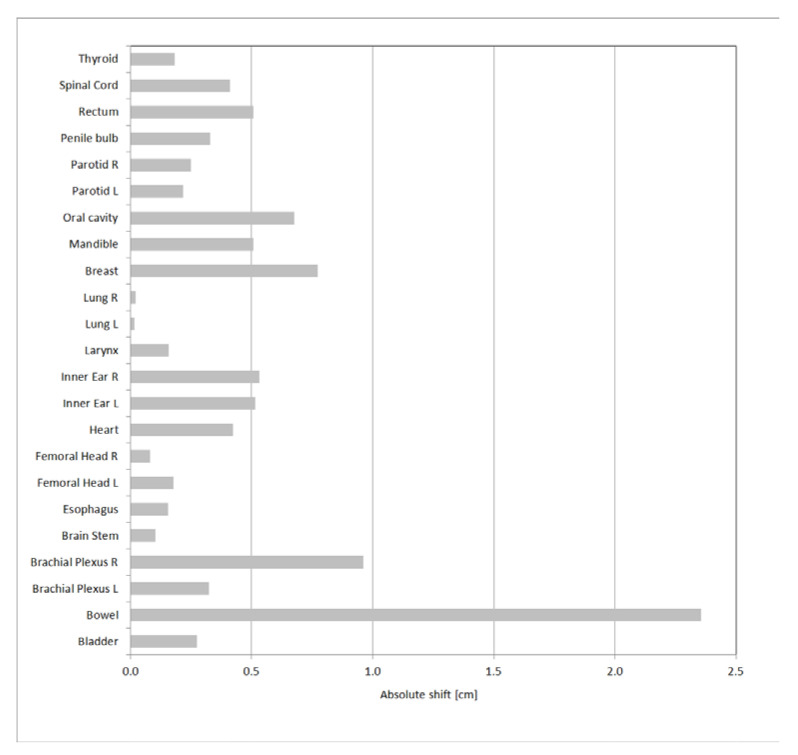
3D absolute shift of the OARs center of mass.

**Figure 6 life-12-02088-f006:**
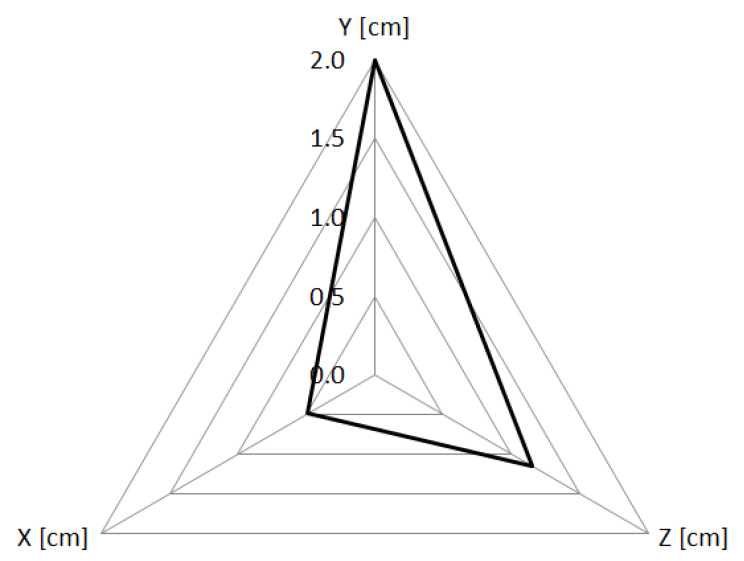
Displacement of the center of mass of the bowel, as absolute value in each direction.

**Table 1 life-12-02088-t001:** Metrics used for DVH. The results of Limbus Contour (LC) and Manual contour (MC) are reported for different treatment sites and OAR ± one standard deviation.

Treatment Site	OAR	DVH Metric	MC Mean ± SD	LC Mean ± SD
H&N	Oral cavity	Mean dose	42.8 ± 4.8	47.9 ± 3.6
	Esophagus	D1cc	44.6 ± 3.4	45.7 ± 5.2
	Larynx	D1cc	49.2 ± 2.4	49.3 ± 1.4
	Larynx	Mean dose	34.8 ± 3.6	35.9 ± 2.1
	Mandible	D1 cc	63.3 ± 6.6	62.9 ± 7
	Spinal Cord	D0.03 cc	28.6 ± 1.7	28.4 ± 1.8
	Spinal Cord	Dmax	29.5 ± 1.3	29.1 ± 1.5
	Inner Ear R	Mean dose	11.3 ± 6.8	10.2 ± 6.2
	Inner Ear L	Mean dose	10.5 ± 6.1	9.3 ± 5.5
	Parotid R	Mean dose	32.4 ± 7.7	36.9 ± 8.7
	Parotid R	V30Gy	48.8 ± 10.9	56.2 ± 11
	Parotid L	Mean dose	29 ± 6.1	33.3 ± 6.6
	Parotid L	V30Gy	43.6 ± 9.8	53 ± 12.4
	Lung R	D30%	5.6 ± 8	5.5 ± 7.9
	Lung L	D30%	4.6 ± 6.4	4.5 ± 6.2
	Thyroid	V45Gy	77.7 ± 25.4	77.7 ± 25.4
	Brain Stem	D0.03cc	35.1 ± 2.9	34.6 ± 2.5
	Brachial Plexus R	D0.03 cc	56.6 ± 1.8	58.7 ± 4.6
	Brachial Plexus L	D0.03 cc	56.8 ± 1.8	58.8 ± 3.7
Left Breast	Lung R	V5Gy	0 ± 0	0 ± 0
	Lung L	V10Gy	12.1 ± 2.3	12.1 ± 2.3
	Lung L	V20Gy	8.5 ± 1.8	8.5 ± 1.8
	Lung L	V5Gy	19.9 ± 3.3	19.9 ± 3.4
	Heart	V25Gy	1.2 ± 1	1 ± 0.9
	Breast	D1cc	0.7 ± 0.1	1.4 ± 0.2
Prostate	Penile bulb	Dmean	14.8 ± 3.2	22.2 ± 15
	Femoral Head R	Dmax	35.3 ± 3	34.9 ± 3
	Femoral Head L	Dmax	37.5 ± 6.1	37.4 ± 5.8
	Rectum	V50Gy	17.4 ± 4.3	19.7 ± 3.6
	Rectum	V60Gy	7.2 ± 2	8.4 ± 1.9
	Rectum	V65Gy	3.8 ± 0.8	4.6 ± 1.7
	Rectum	V68Gy	2 ± 0.7	2.7 ± 1.6
	Bladder	V60Gy	14 ± 2.3	16.6 ± 4.5
Rectum	Femoral Head R	V30Gy	27.6 ± 4.8	27.6 ± 4.5
	Femoral Head R	V40Gy	1.1 ± 0.8	0.8 ± 0.8
	Femoral Head R	V45Gy	0 ± 0	0 ± 0
	Femoral Head L	V30Gy	24.3 ± 9.7	22.7 ± 11.7
	Femoral Head L	V40Gy	0.8 ± 1.3	0.5 ± 0.9
	Femoral Head L	V45Gy	0 ± 0	0 ± 0
	Bladder	V35Gy	30.1 ± 22	32.7 ± 24.8
	Bladder	V40Gy	19.5 ± 16.8	20.1 ± 17.3
	Bladder	V50Gy	0.6 ± 1.1	1.3 ± 1.1
	Bowel	V45Gy	10.4 ± 15.7	289.4 ± 34

## Data Availability

Data are available upon request by contacting the corresponding author.

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
