# Peer review of "Implementation of a Commercial Deep Learning-Based Auto Segmentation Software in Radiotherapy: Evaluation of Effectiveness and Impact on Workflow"

_life, 2022, doi:10.3390/life12122088_

Round 1

Reviewer 1 Report

The authors report a very interesting experience with the application of deep-learning-based software to automate the contouring process.

Considering how volume definition in radiotherapy is often a repetitive and time-wasting process, what the authors report is of absolute scientific interest. In this context, the authors approach the subject with methodological rigor and clarity in reporting their results.

I would like to point out a minor typing error, to correct the surname of the first author of reference 25 on page 10, line 225 (which was correctly stated in the bibliography, but is incorrect in the text)

Author Response

The typographical error has been corrected. Thank you for having noticed it.

Reviewer 2 Report

The authors provided a internal validation of auto-segmentation software also considering the impact on clinical workflow.

Some limitations of the study must be pointed out such as the low number of cases, in this case I suggest to define the total number of contours for each site.

I would also suggest to provide a comparison the results with other similar experience.

An extensive revision of language is necessary.

Author Response

Thank you for your comments.

The number of structures analysed per anatomical region has been explained in section 2.4. Furthermore, the total number of structures was included in the discussions.

The comparison with similar experiences described in literature has been expanded. In particular, comparisons were added for brachial plexuses, bladder and femoral heads. In this way all OARs with notable results, positive or negative, were compared with the literature

Language has been extensively edited

Reviewer 3 Report

Title: Implementation of a commercial deep learning-based auto segmentation software in radiotherapy: evaluation of effectiveness and impact on workflow.

General

This paper investigated clinical investigation in our institution of commercial deep learning-based auto contour software to assess the impact on the radiotherapy workflow with head and neck, prostate, breast, and rectum sites.

The methods and materials of this paper can be said to be a little scientific, but this study seems to have to secure some important points.

1. Did you discover anything new or different from previous research through this study? What do you want to talk about through this research?

2. This study is very similar to previous studies. That is, there is no novelty in this study.

3. The number of patients in the study is too small. The number of patients is too small to explain the purpose of the study with 3 patient cases in 4 sites.

4.page 65~68, it is said that the algorithm model was trained using a public atlase, but it is necessary to write how it was trained in detail. How many patients' data was applied.

5.page 65~68, How was validation performed after trained model?

6.page 67~68, it is necessary to write in detail how this research was conducted.

Author Response

General

This paper investigated clinical investigation in our institution of commercial deep learning-based auto contour software to assess the impact on the radiotherapy workflow with head and neck, prostate, breast, and rectum sites.

The methods and materials of this paper can be said to be a little scientific, but this study seems to have to secure some important points.

  1. Did you discover anything new or different from previous research through this study? What do you want to talk about through this research?
  2. This study is very similar to previous studies. That is, there is no novelty in this study.

1.2.

The study confirms the results presented in literature and extends the geometric evaluations to a greater number of OARs. With regard to time savings, it realistically quantifies the savings thanks to the amount of OARs contoured for each district.

Furthermore, the dosimetric evaluation allows to understand the limits of an unsupervised use of this kind of software.

These results, especially because reported in a single coherent work, represent a novelty compared to other studies present in literature.

In the text, in the discussion section, the novelties and uniqueness of the study have been highlighted.

  1. The number of patients in the study is too small. The number of patients is too small to explain the purpose of the study with 3 patient cases in 4 sites.

We agree that the small number of patients is a limit. However, we have paid attention to the amount of structures analysed per anatomical region  in order to have a realistic assessment of the impact of Limbus Contour on the radiotherapy work flow. This concept has been clarified in the text, underlining the number of structures analysed in total and by disease.

4.page 65~68, it is said that the algorithm model was trained using a public atlase, but it is necessary to write how it was trained in detail. How many patients' data was applied.

This information is described in the work of Wong et al. mentioned in the article. To better explain this concept, the details on the average number of scans  and methods for validation have been explained in the text.

5.page 65~68, How was validation performed after trained model?

We have ask more details about this question to the development manager and we have underlined the answer in the text.

6.page 67~68, it is necessary to write in detail how this research was conducted.

5.6. The manufacturer has not disclosed complete details regarding this software, including neural networks and architecture and validation methods.

Round 2

Reviewer 3 Report

my opinions well considered for this paper.